# Air Pollution/Irritants, Asthma Control, and Health-Related Quality of Life among 9/11-Exposed Individuals with Asthma

**DOI:** 10.3390/ijerph16111924

**Published:** 2019-05-30

**Authors:** Janette Yung, Sukhminder Osahan, Stephen M. Friedman, Jiehui Li, James E. Cone

**Affiliations:** New York City Department of Health and Mental Hygiene, World Trade Center Health Registry, New York, NY 10013, USA; osukhmin@health.nyc.gov (S.O.); sfriedm2@health.nyc.gov (S.M.F.); jli3@health.nyc.gov (J.L.); jcone@health.nyc.gov (J.E.C.)

**Keywords:** 9/11 disaster, asthma, trigger(s), air pollution, irritant(s), health-related quality of life

## Abstract

Asthma control is suboptimal among World Trade Center Health Registry (WTCHR) enrollees. Air pollution/irritants have been reported as the most prevalent trigger among World Trade Center responders. We examined the relationship between air pollution/irritants and asthma control. We also evaluated the association of asthma control with health-related quality of life (HRQoL). We included 6202 enrollees age ≥18 with a history of asthma who completed the WTCHR asthma survey between 2015 and 2016. Based on modified National Asthma Education and Prevention Program criteria, asthma was categorized as controlled, poorly-controlled, or very poorly-controlled. HRQoL indicators include ≥14 unhealthy days, ≥14 activity limitation days, and self-rated general health. We used multinomial logistic regression for asthma control, and unconditional logistic regression for HRQoL, adjusting for covariates. Overall, 27.1% had poorly-controlled and 32.2% had very poorly-controlled asthma. Air pollution/irritants were associated with poorly-controlled (adjusted odds ratio (AOR) = 1.70; 95% CI = 1.45–1.99) and very poorly-controlled asthma (AOR = 2.15; 95% CI = 1.83–2.53). Poor asthma control in turn worsened the HRQoL of asthmatic patients. Very poorly-controlled asthma was significantly associated with ≥14 unhealthy days (AOR = 3.60; 95% CI = 3.02–4.30), ≥14 activity limitation days (AOR = 4.37; 95% CI = 3.48–5.50), and poor/fair general health status (AOR = 4.92; 95% CI = 4.11–5.89). Minimizing World Trade Center (WTC) asthmatic patients’ exposure to air pollution/irritants may improve their disease management and overall well-being.

## 1. Introduction

The prevalence of asthma has increased in the last decade and placed a significant economic burden on the United States and globally. Asthma is a chronic disease characterized by inflammation of the airways, reversible airflow obstruction, and bronchial hyper-responsiveness, with symptoms including wheezing, coughing, tightness of the chest, shortness of breath, and sleep awakening. These symptoms can greatly affect daily activities and quality of life. In the United States, asthma was responsible for $3 billion in losses due to missed work and school days, $29 billion due to asthma-related mortality, and $50.3 billion in medical costs during 2008–2013 [1]. A previous World Trade Center (WTC) study reported that asthma control was poor or very poor in over 68% of World Trade Center Health Registry (WTCHR) enrollees diagnosed after 9/11 [2], likely affecting their quality of life.

The presence of asthma triggers that are difficult to avoid may be responsible for some of this lack of asthma control in many patients. Part of the difficulty in asthma medical management is explicitly defining triggers that contribute to exacerbations [3,4]. A higher number of triggers experienced [3] and a wide array of asthma triggers have been identified and perceived by patients as contributing to the severity and frequency of asthma exacerbations [3,5,6].

Air pollution/irritants are materials in the indoor or outdoor air that can have adverse effects on humans and the ecosystem. These substances can be in the form of particulates, liquid droplets, or gases, and can be of natural origin or man-made. They induce airway inflammation or an allergen-induced response by causing direct cellular injury or by inducing intracellular signaling pathways and transcription factors that are known to be sensitive to oxidative stress [6]. The subsequent mucosal damage and impaired mucociliary clearance may facilitate access of inhaled allergens to the cells of the immune system [6]. Air pollutants/irritants were the most prevalent type of asthma triggers reported in previous studies [7,8]. The most abundant components of air pollution in urban areas are particulate matter (PM), nitrogen dioxide (NO_2_), and ozone (O_3_). PM is a mixture of organic and inorganic solid and liquid particles of different origin and size. NO_2_ can enhance the allergic response to inhaled allergens, and its concentration in ambient air is reportedly associated with cough, wheezing, and shortness of breath in atopic patients. O_3_ is thought to increase asthma morbidity by enhancing airway inflammation and epithelial permeability [6]. High ambient concentrations of these materials are associated with an increased rate of asthma exacerbations [6,9], and were associated with increased post-9/11 asthma-related hospital admissions [9].

Health-related quality of life (HRQoL) has been increasingly recognized as an important endpoint and measure for interventions [10,11,12]. It is regarded as a measure of the effect of the disease on a patient’s life. A previous study reported that greater severity and frequency of exacerbations were significantly associated with decreases in asthma-related quality of life [13]. Poorly-controlled asthma were also found to be related to poorer HRQoL in the United Kingdom [14]. Moreover, asthmatic patients had worse indicators of quality of life compared to the general population regardless of whether symptoms were clinically controlled [15]. 

To better understand the relationship between asthma triggers, asthma control, and HRQoL among asthmatic enrollees who were exposed to the WTC disaster, the current study consists of two parts. The first part aims to supplement existing knowledge regarding the various triggers in an effort to determine whether air pollution/irritants are an important trigger reported by individuals exposed to 9/11. The second part of this study assesses two objectives: (1) to examine the association of self-reported air pollution/irritants trigger with asthma control among all asthmatic enrollees who were exposed to WTC disaster; and (2) to examine the association of different level of asthma control with three domains of impaired HRQoL: self-reported mental or physical unhealthy days, self-reported activity limitation days, and self-reported general health. We hypothesized that reporting air pollution/irritants as an asthma trigger is associated with poorer asthma control, and that poorer asthma control is associated with lower HRQoL among 9/11-exposed individuals with asthma.

## 2. Materials and Methods

The WTCHR has been described in detail elsewhere [16,17]. Briefly, this is a longitudinal cohort study including rescue/recovery workers and volunteers (RRW) and community members not involved in rescue/recovery (lower Manhattan area residents, students, and workers; passersby and commuters on 9/11). Participants were recruited through Lower Manhattan area building or employer lists, or encouraged to enroll via a toll-free telephone number or website [17]. Between 12 September 2003 and 24 November 2004, 71,431 people completed a computer-assisted (95%) or in-person (5%) enrollment interview on demographics, exposures incurred during and after the WTC disaster and health information. Since that baseline enrollment (Wave 1), the Registry has conducted three follow-up surveys (Waves 2 to 4) via mail, website, or telephone interview to collect updated health information. The response rates for Wave 2 (2006–2007), Wave 3 (2011–2012), and Wave 4 (2015–2016) were 65.2%, 60.4%, and 51.6%, respectively. All surveys inquired about enrollees’ medical history and physical and mental health status. The Wave 4 asthma survey began on 3 September 2015 and continued through 20 March 2016, and was administered as a supplement to the Wave 4 Core (Wave 4) survey. We sent out the Wave 4 asthma survey to 14,983 eligible enrollees age ≥18 who reported ever being diagnosed with asthma in any of the three prior wave surveys, either before or after 9/11/2001. Asthma survey data were collected via the internet or through paper mailings and entailed disease management questions, including triggers, asthma control and exacerbation. The Registry’s protocol was approved by the institutional review boards of the US Centers for Disease Control and Prevention and the New York City Department of Health and Mental Hygiene (#02-058).

### 2.1. Study Sample

Of the 14,983 eligible enrollees who were sent the Wave 4 asthma survey, 8482 responded to the survey either fully or partially. Of these, 4653 (55%) were completed by web and 3829 (45%) were completed by paper. In our study, we only included those who reported “yes” to the gate question of “Have you ever been told by a doctor or other health professional that you had asthma?” on the Wave 4 asthma survey (*N* = 7129), though we only mailed the Wave 4 asthma survey to enrollees who had reported an asthma diagnosis on at least one of the three prior Registry surveys (Waves 1 through 3). We excluded enrollees who had missing age (*N* = 3), and those who did not respond to the Wave 4 survey (*N* = 830) since we used the sociodemographic and health variables from Wave 4. We also excluded additional persons who had missing value for the asthma control variable (*N* = 94). The final study sample was 6202 for data analyses.

### 2.2. Study Outcomes and Variables

#### 2.2.1. Determining Air Pollution/Irritants as Trigger

We derived this variable by analyzing asthma triggers following the method of Ritz [4]. Briefly, open-ended responses to the asthma triggers question “Please list up to six of the strongest triggers of your asthma” were grouped into 11 main categories: air pollution/irritants, exercise, infection, allergens (pollen), allergens (animal), allergens (general), air temperature, psychological, allergens (food), medications, and other. Similar to findings in previous studies [7,8], air pollution/irritants was the most prevalent type of trigger, reported by over 50% of the participants. We therefore derived a dichotomous variable of air pollution/irritants.

#### 2.2.2. Asthma Control and HRQoL

We focused on two main outcomes in the second part of the study. The first outcome was asthma control and the second outcome was HRQoL. We categorized asthma control for study participants as having controlled, poorly-controlled, or very poorly-controlled asthma reported in the Wave 4 asthma survey, based on criteria modified from the National Asthma Education and Prevention Program’s Third Expert Panel Report 3 (EPR 3). Briefly, the criteria consist of four components: shortness of breath, wheezing, and/or cough; nighttime awakenings; interference with normal activity; and use of a rescue inhaler or nebulizer. The level of asthma control category was determined by frequency of experience on each component. Participants were assigned to the most severe category in which any component was reported. In our slightly modified criteria, for interference with normal activity, we specified none/a little of the time to well-controlled, some/most of the time to poorly-controlled, all of the time to very poorly-controlled asthma, versus no interference with normal activity, some limitation, and extreme limitation for well-controlled, poorly-controlled, and very poorly-controlled asthma, respectively, based on EPR 3 definition. We also slightly modified use of a rescue inhaler or nebulizer. We specified inhaler use of <3 times/week to well-controlled, 1–2 times/d to poorly-controlled, and >2 times/d to very poorly-controlled asthma, versus ≤2 days/week, >2 days/week, and several times per day for well-controlled, poorly-controlled, and very poorly-controlled asthma, respectively, based on EPR 3 definition [2,18]. In the second analysis, asthma control was treated as predictor variable, and we assessed three dichotomous outcome variables for HRQoL: ≥14 physically or mentally unhealthy days; ≥14 days of poor physical or mental health that keep you from doing your usual activities (activity limitation); and self-rated general health status (excellent, very good, or good versus fair or poor) [19,20].

### 2.3. Covariates

In both analyses, we adjusted for age, gender, race/ethnicity, marital status, education, social integration, smoking status, depression, probable post-traumatic stress disorder (PTSD), gastroesophageal reflux symptoms (GERS), obstructive sleep apnea (OSA), body mass index (BMI), having at least one regular healthcare provider, and WTC dust/debris cloud exposure. WTC dust/debris exposure was a dichotomous variable obtained from the Wave 1 survey. Those who reported having been caught in the dust and debris cloud on 9/11/2001 were considered to have had dust cloud exposure. Comorbid conditions were assessed at Wave 4 survey. The Wave 4 survey began in 20 March 2015 and continued through 31 January 2016, while the Wave 4 asthma survey began on 3 September 2015 and continued through 20 March 2016. The mean time interval between the Wave 4 and Wave 4 asthma surveys was 132 days, with a range from 0 to 323 days. Probable PTSD, depression, and generalized anxiety disorder (GAD) were defined using validated scales. Participants with a score ≥44 on the PTSD Checklist, Stressor-Specific Version, a 17-item scale that inquired about 9/11-related psychological symptoms during the 30 day before Wave 4, were considered to have probable PTSD [21,22]. Depression within the two weeks before completion of Wave 4 was defined as a score ≥10 on the 8-item Patient Health Questionnaire depression scale [23]. Probable GAD during the two weeks before Wave 4 was defined as a score ≥10 on the 7-item GAD scale [24]. Self-reported GERS and history of OSA were also obtained from Wave 4 questionnaire responses. Participants who reported having heartburn at least once a week during the 12 months preceding Wave 4 were considered to have GERS. Smoking history was taken from the most recent questionnaire where smoking data were available. Social integration was defined as either low or high based on participants’ responses to the four questions that assess the components of social integration construct [25]: “In the last 30 days, have you visited, talked, or e-mailed with friends at least twice?”, “In the last 30 days, have you attended a religious service at least twice?”, “In the last 30 days, have you been actively involved in a volunteer organization or club?”, and “About how many close friends or relatives do you have now? Include people you feel at ease with and can talk with about what is on your mind.” Each of these questions received a score of 1 if the response was yes or more than zero. Participants were considered as having lower social integration if the total score was less than two, higher social integration if the total score was two or higher. BMI at Wave 4 was calculated from self-reported height and weight data; a BMI <25 was considered under or normal weight, 25–29.9 was considered overweight and ≥30 was obese.

### 2.4. Statistical Analyses

We used a multinomial logistic regression model to examine the association of self-reported air pollution/irritants as trigger with poor and very poor asthma control, adjusting for covariates. We also used multiple logistic regression model to examine the association between asthma control and three health-related quality of life indicators: ≥14 days of poor physical or mental health, ≥14 days of activity limitation, and self-rated general health, adjusting for the same covariates as analysis 1. In both analyses, we used two-sided tests of significance and assumed a type 1 error of 5%. We used SAS software (Version 9.4, SAS Institute Inc., Cary, NC, USA) for all analyses.

## 3. Results

### 3.1. Asthma Trigger

The frequencies of different self-reported asthma triggers are shown in Figure 1. Half of the WTCHR enrollees with asthma reported air pollution/irritants as their asthma trigger. It was the most prevalent trigger, followed by physical activity and general allergens.

Asthma control status by sociodemographic characteristics, comorbid conditions, reporting air pollution/irritants as an asthma trigger, and WTC dust exposure are shown in Table 1. The overall prevalence of poorly-controlled and very poorly-controlled asthma were 27.1% and 32.2%, respectively. We also observed an increase in prevalence of very poorly-controlled asthma among those with self-reported air pollution/irritants trigger (39.8%), depression (55.8%), and probable PTSD (58.1%).

### 3.2. Self-Reported Air Pollution/Irritants as Trigger and Asthma Control

Enrollees who reported air pollution/irritants as triggers had elevated odds ratios for poorly-controlled ((adjusted odds ratio (AOR): 1.70, 95% CI: 1.45–1.99) and very poorly-controlled asthma (AOR: 2.15, 95% CI: 1.83–2.53), after adjusting for covariates (Table 2).

### 3.3. Asthma Control and HRQoL

All three HRQoL indicators were strongly associated with asthma control level (Table 3). Enrollees with poorly-controlled asthma had higher odds ratios for ≥14 physically or mentally unhealthy days (AOR: 2.12, 95% CI: 1.78–2.52), ≥14 days of activity limitation (AOR: 2.17, 95% CI: 1.70–2.77), and self-rated poor general heath (AOR: 2.66, 95% CI: 2.22–3.18), compared to those with well-controlled asthma. Stronger associations were observed for very poorly-controlled asthma.

### 3.4. Mental Health, Asthma Control and HRQoL

Consistent with previous WTCHR findings, both depression and PTSD were associated with asthma control level [2]. Enrollees who reported depression or PTSD had higher odds ratios for poorly-controlled (AOR: 1.46, 95% CI: 1.14–1.87; AOR: 2.64, 95% CI: 2.04–3.43) and very poorly-controlled asthma (AOR: 2.32, 95% CI: 1.83–2.94; AOR 3.39, 95% CI: 2.63–4.37), respectively. Moreover, both mental illnesses were associated with ≥14 physically or mentally unhealthy days (AOR: 5.24, 95% CI: 4.22–6.51; AOR 3.08, 95% CI: 2.46–3.86); ≥14 days of activity limitation (AOR: 3.18, 95% CI: 2.58–3.92; AOR: 2.13, 95% CI: 1.72–2.64); and self-rated poor general heath (AOR: 2.65, 95% CI: 2.17–3.24; AOR: 2.04, 95% CI: 1.66–2.51) for depression and PTSD, respectively.

## 4. Discussion

### 4.1. Main Findings

More than half of WTC Health Registry enrollees who have asthma had poorly- or very poorly-controlled symptoms. Consistent with other studies in various populations [7,8], our study found that air pollution/irritants were the most prevalent asthma trigger, and poor asthma control worsened the HRQoL of asthmatic patients. Additionally, our study found that those who identified air pollution/irritants as their asthma trigger were more likely to have poorly- or very poorly-controlled asthma than those who did not report air pollution/irritants as their trigger. Our findings underscore the importance of reduction in indoor and outdoor air pollution exposure, in combination with optimal asthma control, to improve HRQoL among those who were exposed to 9/11 disaster.

### 4.2. Air Pollution/Irritants and Asthma Control

Consistent with existing knowledge, our study supports the premise that air pollution/irritants are significantly associated with poorer asthma control [5,7,9]. This relationship persists independent of other possible and known risk factors, including 9/11 dust cloud exposure. Higher pollutant levels such as increased daily NO_2_ and O_3_ have been associated with increased asthma-related emergency department visits after the WTC attacks [9]. Air pollutants may not only increase the frequency and intensity of symptoms but may also promote airway sensitization to airborne allergens in predisposed persons [6]. WTC-dust exposed asthmatic patients may be likely to have poorer asthma control when exposed to certain environmental triggers that recall the WTC attack, such as air pollution/irritants.

Measures reducing indoor and outdoor air pollution/irritants exposure, such as high-efficiency particulate air filtration system, can improve indoor air quality by reducing levels of PM_2.5_ and particle count [26,27]. In addition, educating enrollees about trigger identification and avoidance may prevent exacerbations and help to improve overall asthma management [13,28,29,30]. Other sociodemographic factors, including older age, lower education level, and higher BMI also contributed to poorer asthma control, consistent with existing literature [2,10,30,31,32]. This highlights the possible prevention of asthma exacerbation through lifestyle modification focusing on maintaining normal BMI among asthmatic persons.

### 4.3. Asthma Control and HRQoL

Our findings are also congruent with the existing literature that poorer asthma control is associated with worse HRQoL [10,14,33]. This association continued to be significant after adjusting for known covariates, suggesting that perceived well-being is a complex interplay between the pathophysiological manifestations of asthma, socio-demographic factors, and physical and mental health comorbidities. A previous study also suggested that increased severity of asthma was linked to lower HRQoL among the poorly controlled [33]. In addition, people with poorly controlled asthma and poor HRQoL are more likely to have an asthma attack or been admitted to hospital more than once in the previous 12 month [14]. Since the frequency of asthma attack and asthma-related hospitalization are long-term outcome indicators for asthma [34,35], future studies looking at these outcomes may be helpful to understand how different level of asthma control affects the overall disease outcome.

### 4.4. Mental Health, Air Pollution/Irritants, and Asthma Control

Our study also supported previous findings that mental health is an important factor for asthma severity, exacerbation and HRQoL [7,15,30,36]. A previous study showed that the effect of air pollution/irritants on asthma control was heightened among WTC workers with panic disorder and PTSD [8]. The air pollution trigger was also found to be associated with anxiety in another study [36], suggesting those with comorbid anxiety might be more susceptible to exacerbation when exposed to air pollution/irritants. Moreover, anxiety and depression were reported to be associated with lower HRQoL among asthmatic patients [15,30,32,36]. Mental illness constitutes a substantial component of asthma disease manifestation. It is necessary to include mental health management in any control measures targeted to improve disease outcome and HRQoL among asthmatic patients.

### 4.5. Strength and Limitations

The data in this study were self-reported, and we did not have quantitative assessment of asthma control, which requires clinical measurement of pulmonary function such as peak expiratory flow (PEF) and forced expiratory volume (FEV). However, such data may not be essential to our analysis since spirometric lung function may be largely unrelated to perceived asthma triggers [36], though it might support the reliability of our asthma control outcome. A strength of this study was that it included a large sample size and data were available on a wide range of comorbid health conditions. There is a high level of agreement on risk factors of asthma control with previous studies of WTC-exposed populations [2,8], supporting the reliability of our findings.

The asthma triggers information in this study was also self-reported. Given the subjectivity of trigger perception and one’s physiological response to it, the validity of self-reported air pollution/irritants was in question. There are publicly available data on air quality from the air monitoring stations, such as the air quality index from the Environment Protection Agency, which provides information on level of specific pollutants such as O_3_, sulfur dioxide or NO_2_ by geographic area. Future studies integrating such local monitoring environmental data [37,38] with patients’ geospatial information such as residential address, may be helpful to validate the accuracy of self-reported air pollution/irritants as trigger.

Given the complexity of treatment regimens and our lack of clinical information on enrollees, we did not account for medication use and treatment adherence, which are both known to play an important role in asthma severity and control [39]. Since self-rated severity of disease has been associated with HRQoL or asthma-related quality of life [10,13,30,33,35,36], future studies that consolidate medication and treatment adherence information to assess asthma severity maybe useful to better understand their role in the relationship of asthma triggers, severity and control, as well as between asthma triggers and HRQoL.

## 5. Conclusions

Asthma control is closely associated with the perception of asthma trigger exposure, especially to air pollution. Awareness of appropriate preventive measures against air pollution/irritants exposure, and ensuring compliance with these measures may improve asthma control. Asthma control is also closely associated with HRQoL. Optimal asthma control, in combination with treatment for mental health conditions and lifestyle modification advice such as maintaining healthy weight by encouraging diet modification and physical activity, may subsequently lead to improved HRQoL for asthmatic persons exposed to the WTC disaster.

## Figures and Tables

**Figure 1 ijerph-16-01924-f001:**
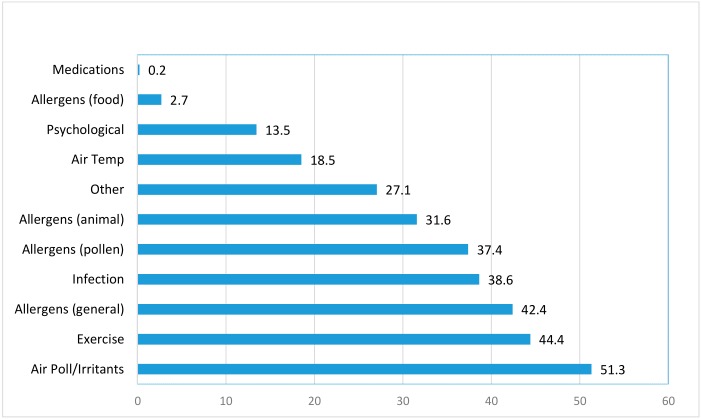
Percentage of each self-reported asthma trigger among World Trade Center Health Registry enrollees with asthma (not mutually exclusive).

**Table 1 ijerph-16-01924-t001:** Prevalence of asthma control by sociodemographic, air pollution/irritants, and World Trade Center (WTC) dust cloud exposure among 9/11-exposed individuals with asthma (*N* = 6202).

Characteristics at Wave 4 *	Total *	% Controlled	% Poorly-Controlled	% Very Poorly-Controlled
*Gender*				
Male	3357	38.2	26.2	35.6
Female	2845	43.6	28.2	28.2
*Age*				
<45	1026	59.2	22.2	18.6
45–64	3897	38.0	27.8	34.2
≥65	1279	33.9	29.2	36.9
*Race/Ethnicity*				
White non-Hispanic	4109	44.3	25.0	30.7
Black non-Hispanic	622	31.4	33.6	35.1
Hispanic	938	29.5	33.5	37.0
Asian	286	46.9	25.9	27.3
Other	247	38.5	24.3	37.3
*Marital Status*				
Never married	961	44.8	27.4	27.9
Married/living with a partner	3978	42.6	26.6	30.8
Widowed, divorced, or separated	1163	32.0	28.1	39.9
*Education*				
College/post-graduate	3194	52.5	24.1	23.5
Some college	1964	31.7	29.7	38.7
High school or less	970	20.3	31.4	48.3
*Smoking Status*				
Never smoker	3832	44.0	26.3	29.7
Previous smoker	1791	39.1	26.8	34.1
Current smoker	438	24.2	31.3	44.5
*BMI*				
Normal (<25)	1450	54.6	22.7	22.8
Overweight (25–29)	2111	41.8	27.5	30.7
Obese (>30)	2436	31.7	29.2	39.2
*Social Integration*				
High	5587	42.8	26.9	30.3
Low	411	16.8	28.7	54.5
*Depression*				
No	4384	50.2	25.9	23.9
Yes	1452	15.8	28.4	55.8
*Probable PTSD*				
No	4632	50.0	25.9	24.1
Yes	1452	12.4	29.5	58.1
*GERS*				
No	3450	50.7	25.1	24.2
Yes	2344	27.8	29.0	43.2
*Sleep Apnea*				
No	4081	49.7	25.6	24.7
Yes	1647	21.9	28.5	49.5
*Had at least one regular healthcare provider*				
No	415	42.4	28.2	29.4
Yes	5681	40.7	26.9	32.4
*Air pollution/irritants as an important trigger*				
No	2777	49.2	24.8	26.0
Yes	2926	30.3	29.9	39.8
*Dust cloud exposure at 9/11*				
No	2587	46.9	25.3	27.8
Yes	3592	36.1	28.5	35.4

* Variable may not add up to total due to missing category. BMI: body mass index; PTSD: post-traumatic stress disorder. GERS: gastroesophageal reflux symptoms.

**Table 2 ijerph-16-01924-t002:** Multivariable odds ratios for association of level of asthma control with air pollution/irritants, controlling for selected sociodemographic factors, WTC dust cloud exposure, physical and mental health co-morbidities among 9/11-exposed individuals with asthma, 2015–2016.

Characteristics at Wave 4 *	Poorly-Controlled	Very Poorly-Controlled
	Adjusted OR (95% CI)	Adjusted OR (95% CI)
*Gender*		
Female	referent	referent
Male	1.08 (0.91–1.28)	1.52 (1.27–1.82)
*Age*		
<45	referent	referent
45–64	1.41 (1.13–1.75)	1.53 (1.21–1.94)
≥65	1.78 (1.36–2.33)	2.13 (1.60–2.83)
*Education*		
College/post-graduate	referent	referent
Some college	1.44 (1.20–1.71)	1.79 (1.50–2.15)
High school or less	1.80 (1.39–2.33)	2.76 (2.14–3.55)
*Smoking Status*		
Never smoker	referent	referent
Former smoker	1.07 (0.90–1.27)	1.19 (1.00–1.42)
Current smoker	2.11 (1.50–2.97)	2.61 (1.86–3.67)
*BMI*		
Normal (<25)	referent	referent
Overweight (25–29)	1.32 (1.08–1.63)	1.22 (0.98–1.52)
Obese (>30)	1.58 (1.28–1.95)	1.68 (1.35–2.10)
*Social Integration*		
High	referent	referent
Low	1.60 (1.09–2.37)	1.77 (1.21–2.60)
*Depression*		
No	referent	referent
Yes	1.46 (1.14–1.87)	2.32 (1.83–2.94)
*Probable PTSD*		
No	referent	referent
Yes	2.64 (2.04–3.43)	3.39 (2.63–4.37)
*GERS*		
No	referent	referent
Yes	1.47 (1.24–1.73)	1.72 (1.45–2.03)
*Sleep Apnea*		
No	referent	referent
Yes	1.38 (1.13–1.68)	1.78 (1.46–2.16)
*Dust cloud exposure at 9/11*		
No	referent	referent
Yes	1.22 (1.05–1.43)	1.36 (1.16–1.59)
*Air pollution/irritants*		
No	referent	referent
Yes	1.70 (1.45–1.99)	2.15 (1.83–2.53)

* Non-significant covariates were not reported in the table.

**Table 3 ijerph-16-01924-t003:** Multivariable odds ratios for association of three health-related quality of life indicators with asthma control, controlling for selected sociodemographic factors, WTC dust cloud exposure, physical and mental health co-morbidities among 9/11-exposed individuals with asthma, 2015–2016.

Characteristics at Wave 4 *	≥14 Days of Poor Physical or Mental Health	≥14 Days of Activity Limitation	Fair/Poor General Health
	Adjusted OR (95% CI)	Adjusted OR (95% CI)	Adjusted OR (95% CI)
*Sex*			
Female	referent	referent	referent
Male	0.81 (0.70–0.95)	1.12 (0.92–1.35)	1.16 (0.99–1.37)
*Age*			
<45	referent	referent	referent
45–64	1.26 (1.03–1.55)	1.35 (1.03–1.77)	1.58 (1.28–1.97)
≥65	1.33 (1.03–1.71)	1.92 (1.40–2.63)	1.76 (1.36–2.29)
*Race/ethnicity*			
White non-Hispanic	referent	referent	referent
Black non-Hispanic	1.05 (0.81–1.35)	1.37 (1.03–1.81)	1.32 (1.03–1.70)
Hispanic	1.13 (0.91–1.40)	1.05 (0.83–1.33)	1.40 (1.13–1.73)
Asian	1.10 (0.77–1.59)	0.76 (0.46–1.25)	2.46 (1.72–3.52)
Other	1.11 (0.76–1.63)	1.32 (0.87–2.01)	1.48 (1.01–2.17)
*Marital Status*			
Never married	referent	referent	referent
Married/living with a partner	0.87 (0.71–1.07)	0.71 (0.55–0.90)	0.78 (0.64–0.97)
Widowed, divorced, or separated	1.11 (0.87–1.43)	0.93 (0.71–1.23)	0.91 (0.71–1.17)
*Highest Education Attainment*			
College/post-graduate	referent	referent	referent
Some college	1.29 (1.09–1.51)	1.47 (1.22–1.78)	1.40 (1.19–1.65)
High school or less	1.47 (1.18–1.84)	1.94 (1.53–2.46)	1.90 (1.53–2.37)
*BMI*			
Normal (<25)	referent	referent	referent
Overweight (25–29)	1.09 (0.90–1.32)	0.98 (0.77–1.25)	1.29 (1.05–1.58)
Obese (>30)	1.17 (0.96–1.42)	1.11 (0.87–1.41)	1.50 (1.22–1.83)
*Social Integration*			
High	referent	referent	referent
Low	1.80 (1.25–2.59)	1.51 (1.12–2.04)	1.95 (1.40–2.70)
*Depression*			
No	referent	referent	referent
Yes	5.24 (4.22–6.51)	3.18 (2.58–3.92)	2.65 (2.17–3.24)
*Probable PTSD*			
No	referent	referent	referent
Yes	3.08 (2.46–3.86)	2.13 (1.72–2.64)	2.04 (1.66–2.51)
*GERS*			
No	referent	referent	referent
Yes	1.46 (1.25–1.70)	1.57 (1.32–1.87)	1.72 (1.48–2.00)
*Sleep Apnea*			
No	referent	referent	referent
Yes	1.55 (1.30–1.85)	1.52 (1.25–1.83)	1.81 (1.53–2.14)
*Asthma Control*			
Controlled	referent	referent	referent
Poorly-controlled	2.12 (1.78–2.52)	2.17 (1.70–2.77)	2.66 (2.22–3.18)
Very poorly-controlled	3.60 (3.02–4.30)	4.37 (3.48–5.50)	4.92 (4.11–5.89)

* Non-significant covariates were not reported in the table.

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
