# Peer review of "Air Pollution/Irritants, Asthma Control, and Health-Related Quality of Life among 9/11-Exposed Individuals with Asthma"

_ijerph, 2019, doi:10.3390/ijerph16111924_

Round 1
Reviewer 1 Report
In their manuscript “Air pollution/irritants, asthma control and health-related quality of life among 9/11-exposed individuals with asthma” the authors investigated the association of:
1. self-reported air pollution/irritants trigger with asthma control clinical;
2. different levels of asthma control with three domains of impaired HRQoL: self-reported mental or physical unhealthy days, self-reported activity limitation days, and self-reported general health.
in a cohort exposed to the particular matter generated during the World Trade Center incident in 9/11/2001.
The thematic of research is well introduced and the results nicely discussed. However, I have some remarks considering the abstract and methods.
Regarding the abstract, please improve the distinction between the purpose of the research and the methods by avoiding repetitions in the words used. The main results of the two research objectives should appear clearly, including the association asthma - HRQoL.
Regarding the methods, I am quite puzzled by the tool used by the authors to quantify the level of social support. This tool doesn’t look to be validated in social science, as is the case for example for the Multidimensional Scale of Perceived social support. To validate this tool, I will suggest to remove from the present manuscript the association between asthma and social support and to publish it elsewhere, in a sociological journal.
Small remark:
Mention please the significance of the acronym PTSD the first time that you cite it in the text (for example in line 146).
Author Response
Dear reviewer,
Please see attached pdf file for point-by-point response to your comments, for the manuscript titled "Air pollution/irritants, asthma control and health-related quality of life among 9/11-exposed individuals with asthma". All your comments have been addressed and by red color font.
Thank you very much for taking your time to review my manuscript and the responses to reviewer document. I look forward to hearing from you soon!
-Janette

Reviewer 2 Report
Did you have a hypothesis?
If so please state and discuss as such.
Tables spread across two pages. This makes it difficult to take in the data.
How is this paper different that what is already known about asthma?
How will this work inform future research?
Author Response

(The authors gave the same response as above.)

Round 2
Reviewer 1 Report
The authors responded wisely to all my remarks.
I do not have any additional remarks to make on the new version.